# Influence of Urbanization and Foreign Direct Investment on Carbon Emission Efficiency: Evidence from Urban Clusters in the Yangtze River Economic Belt

## Shijian Wu * and Kaili Zhang

School of Economics and Management, Shandong University of Science and Technology, Qingdao 266590, China; zkl970211@163.com

\* Correspondence: wsj@sdust.edu.cn; Tel.: +86-135-8933-8136

**Abstract:** Reducing carbon emissions and realizing green, circular, and low-carbon development is essential for high-quality economic development. Following the construction of a superefficiency SBM model and combining the panel data of three major urban agglomerations in the Yangtze River Economic Belt from 2003 to 2017, carbon emission efficiency was measured and analyzed. A spatial Durbin model (SDM) was incorporated to analyze the urban agglomerations in the Yangtze River Economic Belt and the impact of urbanization quality and foreign direct investment (FDI) on carbon emission efficiency. Finally, the SDM model was used to decompose the spillover effect. Generally, carbon emission efficiency in the three major urban agglomerations in the Yangtze River Economic Belt is low, with regional differences. FDI only has a positive impact on the carbon emissions of the Yangtze River Delta and the middle reaches of the Yangtze River. Furthermore, urbanization and population density have led to high levels of carbon emission in the region; however, the industrial structure and energy intensity factors have inhibited the improvement of regional carbon emission efficiency. Improving the quality of urbanization and trade structure is important to achieve energy conservation and emission reductions, which are pillars of sustainable economic development.

**Keywords:** carbon emission efficiency; foreign direct investment (FDI); spatial Durbin model; urbanization; Yangtze River Economic Belt

---





## 1. Introduction

In recent years, high amounts of greenhouse gases (GHG), such as carbon dioxide ($CO_2$), have been emitted, causing global warming and increasing the frequency of extreme weather events and natural disasters, which pose serious threats to ecosystems inhabited and exploited by humans, as well as the resources therein. The Yangtze River Economic Belt spans the three major regions of China, including the east, central, and west regions. The economic belt has the highest economic potential and the greatest strategic government support outside the open coastal areas. The Yangtze River Economic Belt also facilitates interaction and cooperation among the east and the west, coordinating and promoting the opening of coastal cities and rivers. It plays a crucial role in the new round of reform and opening and is called the "backbone of China's economy" [1]. The Yangtze River Economic Belt is only approximately 20% of the land area of China but supports more than 45% of the total economic output and hosts more than 40% of the Chinese population. Regardless of the geographical location, economic foundation, natural environment, or resource level, it has an irreplaceable strategic significance for China's economic development. With the continuous acceleration of urbanization and industrialization in recent years, the Yangtze River Economic Zone has attracted massive foreign direct investment (FDI). By the end of 2019, the foreign capital hosted in the Yangtze River Economic Belt was 368.3 billion yuan, an increase of 8% from the previous year. The national proportion is as high as 49%, making it China's largest foreign investment area. Conversely, the conflicts between the

low threshold of foreign investment, the extensive promotion of urbanization, resource carrying capacity, and the environment have become increasingly prominent, with serious environmental problems in the cities of the Yangtze River Economic Belt. Therefore, an in-depth study of the relationship between urbanization, FDI, and carbon emission efficiency in the Yangtze River Economic Belt urban clusters could facilitate the formulation of rational and effective carbon emission reduction measures and the realization of low-carbon and green development in the region. In the context of frequent multinational investment and increasing calls for environmental protection, scholars have gradually realized that environmental pollution shifts from developed countries or regions to developing countries through FDI, and the potential influence of FDI on carbon emissions has begun to emerge. Different perspectives exist with regard to the influence of FDI on environmental pollution. One view is that FDI forms a "pollution refuge." To circumvent the strict environmental regulations of developed countries and minimize environmental pollution in the regions, foreign pollution-intensive multinational companies have transferred polluting companies to countries or regions with relatively lax environmental controls so that they have "pollution refuges" [2–4].

Markusen et al. observed that developed countries save environmental governance costs by transferring pollution-intensive industries or production links to developing countries [5,6]. This degrades the local environments and intensifies the pressure on the host country to reduce carbon emissions. In addition, according to Lin et al., when environmental pollution and FDI have marked positive spatial correlation [7], the high FDI values are generally in high $CO_2$-emission areas, and as the scope of FDI continues to expand, carbon emissions gradually shift to relatively less-developed areas [8,9]. Zakarya and To et al. explored the relationship between FDI and carbon emissions based on data from different countries and reported that the introduction of foreign capital would increase $CO_2$ emissions [10,11], and Xie and Shahbaz et al. confirmed the "pollution refuge" hypothesis [12,13]. Another view is that FDI has a "carbon halo" effect, which would not deteriorate the environmental quality of the host country through the introduction of measures such as clean technologies, improvement of production efficiency, and the transfer of environmental management experience. Therefore, FDI would facilitate the improvement of regional environmental management practices and reduce local $CO_2$ emissions [14–16]. Zhu et al. studied the relationship between FDI and carbon emissions in five member states of the Association of Southeast Asian Nations based on the panel quantile regression model and observed a negative influence of FDI on carbon emissions [17]. Zang et al. used a time-series regression model to analyze the panel data of the logistics industry in China and reported that the introduction of foreign capital had reduced the $CO_2$ emissions in the industry and reduced carbon emission intensity to a certain degree [18]. In addition, Liu et al. reported that FDI improved the pollution discharge technologies used by local enterprises, following an investigation of the interaction between FDI and environmental quality [19], which facilitates the reduction of $CO_2$ emissions [20]. There persists a minority view today that the impact of FDI on carbon emissions is linked to numerous factors such as environmental regulations, governance technologies, economic development levels in countries, human-capital structure, and domestic enterprises' absorptive capacity. Furthermore, there are differences across regions, in the form of the "scale effect" [21,22]. Based on the time-series data of Shandong Province from 2000 to 2016, Wang and Chu studied the mechanism of FDI on energy consumption in Shandong Province according to three aspects: scale effect, structure effect, and technology effect. The results showed that FDI had negative scale effect, structural effect, and positive technical effect on energy consumption in Shandong Province, and the total effect was negative [23]. Li et al. conducted an empirical analysis of the threshold effect of FDI on environmental quality based on the threshold regression method. They found that under high per capita income, high human capital, or high environmental regulation, the introduction of FDI facilitates the improvement of local environmental governance, while under low levels of human capital and environmental governance, the effect is the opposite [24]. Liu et al. also observed that FDI

and environmental pollution had a strong positive correlation in space based on a spatial econometric model [25]. On the contrary, Zhou et al. found that China's urban carbon emissions did not follow the inverted U-shaped hypothesis of the traditional EKC curve theory but proposed an inverted N-shaped. In addition, due to "hidden trade carbon," current FDI has increased the carbon emissions of Chinese cities. However, during the first phase of the lag, it greatly reduced the city's carbon emissions [26]. Areas with high FDI agglomeration have high pollution agglomeration. In short, the influence of FDI on carbon emissions remains highly uncertain, and the influence of FDI on carbon emissions may vary due to differences in investment motivations in various countries. As the most economically active area in the Yangtze River Economic Belt and the area with the greatest use of foreign capital, urban agglomerations in the region should reflect international environmental protection trends.

In addition to FDI being the major factor influencing $CO_2$ emissions, numerous researchers have explored the influence of urbanization and other factors. Based on panel data from 99 countries, Poumanyvong et al. reported that urbanization is associated with increased $CO_2$ emissions [27]. With an increase in per capita income, carbon emissions increased annually. Kasman et al. arrived at different conclusions based on a comparison between the level of urbanization and $CO_2$ in the EU and other developed countries and the BRICS [28], and other relatively underdeveloped regions [29,30]. Ali et al. used auto-regressive distribution lag (ARDL) to analyze the impact of urbanization on Pakistan's carbon dioxide emissions. The results show that urbanization increases carbon emissions in both the long and short term [31]. Wang came to the same conclusion based on an analysis of 137 countries [32]. In China, the impact of emissions varies with the stage of development; since the reform and opening-up, China's urbanization rate has continued to rise, and urbanization has increased from 17.92% in 1978 to 60.60% in 2019 [33]. Regarding the relationship between urbanization and $CO_2$ emissions, several scholars believe that the two are generally positive. Similarly, Lin et al. demonstrated that urbanization promotes carbon emissions significantly [34]. Xu et al. also observed that urbanization in China increased carbon emissions and that the process of urbanization would continue to increase carbon emissions [35]. Guo et al. also confirmed a positive relationship between the two factors; conversely, several scholars believe that urbanization could reduce carbon emissions [36]. Zhao et al. used co-integration and Granger-causality testing methods and found that urbanization had considerable adverse impacts on $CO_2$ emissions [37]. Niu used Chinese provincial panel data from 2002 to 2016 for analysis and found that the increase in urbanization rate had an overall inhibitory effect on the growth of carbon emissions [38]. Lu also demonstrated that, overall, urbanization, especially in the Midwest, would facilitate the achievement of carbon emission reductions in China. In addition, several scholars contend that with the acceleration of urbanization, the relationship between urbanization, and $CO_2$ emissions would be nonlinear, and the impact of urbanization on carbon emissions varies regionally [39]. Based on dynamic panel data for each province in China from 1995 to 2012, Li observed that urbanization had an inverted U-shaped relationship with carbon emissions [40]. Based on the KAYA identity, Yu et al. observed that urbanization in the Beijing–Tianjin–Hebei region also had an inverted "U"-shaped impact on carbon emissions, which enhanced emissions before suppressing them [41]. Xu et al. also reached a similar conclusion on the impact of urbanization on carbon emissions in the Pearl River Delta region [42]. Wang employed the STIRPAT model to conduct an empirical analysis on panel data of 29 provinces in China and demonstrated that the relationship between urbanization and carbon emissions is an "N" or an inverted "N" curve [43]. Sun et al. observed that in the early stages of urbanization, the indirect impact of urbanization on carbon emissions was not significant [44]. However, with urbanization, the emission of $CO_2$ and other GHGs increases significantly, and the growth rate of carbon emissions gradually decreases in the later stages of urbanization [45,46]. Shi et al. used the expanded STIRPAT model, according to the level of urbanization, and studied the impact of the urbanization rate on the carbon emissions of different urbanized areas [47]. They reported that the level

of urbanization in the first - and second-tier regions would increase carbon emissions, while the increase in the level of urbanization in the third-tier regions would reduce carbon emissions. The differences in stages of development in different regions leads to different urbanization characteristics. Wang and Li divided China's 29 provinces into three categories based on the quality of urbanization and analyzed the impact of six urbanization quality indicators on carbon emissions. The survey results showed that the impact of different factors on carbon emissions varies greatly among provinces [48]. Therefore, it is necessary to study the relationship between urbanization and carbon emissions through in-depth studies conducted across regions and cities. The conclusions drawn from such studies could guide interventions nationally and locally.

In summary, the academic research on FDI and urbanization development on carbon emissions has produced valuable results which provide direction for the in-depth development of this research. However, existing research mainly focuses on national or provincial areas, and there is insufficient literature studying the perspective of urban agglomerations. In addition, the quality of urbanization measured by a single population urbanization rate has a certain degree of uncertainty, and it is difficult to truly and fully reflect China's new urbanization. Existing research also ignores the spatial connections between cities in different regions; FDI and urbanization can have spatial spillover and radiation effects on carbon emissions, but existing studies focus on the direct impact on the efficiency of regional carbon emissions. Therefore, this study takes the Yangtze River Delta urban agglomeration, the middle reaches of the Yangtze River, and the Chengdu–Chongqing urban agglomeration as the research objects and uses the superefficiency relaxation measure (SBM) based on data from 2003 to 2017 to measure the carbon emission efficiency levels of the three urban agglomerations and internal cities. On this basis, this study refines the quality analysis of urbanization and constructs a spatial Durbin model (SDM) for modeling and understanding economically related FDI and urbanization. A comprehensive analysis of key factors is carried out to conduct an in-depth analysis of the spatial spillover effects of carbon emission efficiency in each urban agglomeration, with a view for providing a certain theoretical basis and support for the region to rationally attract investment and achieve green, low-carbon sustainable development.

## 2. Materials and Methods

### 2.1. Urban Agglomeration Status in the Yangtze River Delta

The Yangtze River Delta, the middle reaches of the Yangtze River, and the Chengdu–Chongqing urban agglomeration are all national-level urban clusters located in the upper, middle, and lower reaches of the Yangtze River, corresponding to the eastern, central, and western regions of China, which are typical and representative. Since the "Eleventh Five-Year Plan," China has successively issued a series of targeted development strategies such as the "Guiding Opinions on Relying on the Golden Waterway to Promote the Development of the Yangtze River Economic Belt" to further plan and adjust the development plans of the three major urban agglomerations. It provides strong support for the sustainable integrated development of urban clusters and surrounding cities [49]. Because of this, this study conducts an empirical study based on panel data of 69 cities in the three major urban clusters during the 2003–2017 sample period (Table 1). At the end of 2017, the total population of the three urban agglomerations was 370 million, accounting for 27% of the national population, and the population density was 518.58 people/km$^2$ [50]. In 2017, the total GDP of the three urban agglomerations was 29.54 trillion yuan. It accounts for 35.71% of the national GDP; the land area is 713,700 km$^2$, accounting for 7.4% of the national land area.

### 2.2. Superefficiency SBM Model

The data envelopment analysis (DEA) method was first put forward by Charnes in 1978. Since DEA does not need to estimate parameters in advance, nor does it need to make weight assumptions, it directly calculates the input–output efficiency of decision-

making units through the ratio of the weighted sum between output and input, which is widely used in resource and environmental efficiency evaluation. To incorporate input, output, and pollution into the measurement framework, Tone established a new DEA model, the SBM model [51]. By setting up an output slack variable to evaluate decision-making units with undesired output, it addresses the fundamental input–output issues and can accurately measure the level of efficiency. However, in the SBM–DEA model measurement results, the efficiency value of multiple decision-making units is often equal to 1, and distinguishing the effective decision-making units between the SBM model and the traditional DEA model is challenging. Andersen et al. [52] proposed a method of further distinguishing effective decision-making unit (DMU), which is known as spatial error model (SEM). In the superefficiency model, the efficiency frontier of a DMU is composed of DMUs other than the evaluated DMU. When the evaluated DMU is outside the frontier, the efficiency value is greater than 1, and the efficiency value of DMU can be effectively distinguished. Based on the SBM model, Tone further defines the superefficiency SBM model [51], which is a model combining the superefficiency DEA model and the SBM model. By integrating the advantages of the two models, the superefficiency SBM model can identify the efficient DMU at the leading edge.

**Table 1.** Basic data on urban agglomerations (2017).

| Urban Agglomeration | Number of Cities | Land Area (10,000 km$^2$) | GDP (100 Million yuan) | Population (10,000) | Population Density (Person/km$^2$) | Output Density (10,000/km$^2$) |
|---|---|---|---|---|---|---|
| Yangtze River Delta | 26 | 21.17 | 165,193.65 | 13,089 | 618.2805 | 7803.1955 |
| Middle reaches of the Yangtze River | 28 | 31.7 | 76,838.4104 | 13,082 | 412.6813 | 2423.9246 |
| Chengdu–Chongqing | 15 | 18.5 | 53,359.78 | 10,840 | 585.9459 | 2884.3124 |
| Sum | 69 | 71.37 | 297,391.84 | 37,011 | 518.58 | 4166.9 |

Therefore, here, the authors assume that the income is constant, and considering actual carbon emissions, a superefficiency SBM model with an undesired output measures the comprehensive efficiency of energy conservation and emission reduction in 69 cities in the three major urban clusters of the Yangtze River Economic Belt. Suppose there are $n$ decision-making units: each decision-making unit has $m$ types of input, $r_1$ types of expected output, and $r_2$ types of undesired output. The vectors $x, y^d$ and $y^u$ represent input, expected output, and undesired output, respectively. Among them, $x \in R^m, y^d \in R^{r_1}, y^u \in R^{r_2}$, the definition matrix $X, Y^d$, and $Y^u$ are: $X = [x_1 \dots x_n] \in R^{m \times n}, Y^d = \left[ y_1^d \dots y_n^d \right] \in R^{r_1 \times n}$, and $Y^u = \left[ y_1^u \dots y_n^u \right] \in R^{r_2 \times n} Y^u = \left[ y_1^u \dots y_n^u \right] \in R^{r_2 \times n}$. Following this, the form of the superefficiency SBM model is:

$$\min\theta = \frac{\frac{1}{m} \sum\limits_{i=1}^{m} (\bar{x}/x_{ik})}{\frac{1}{r_1+r_2} \left( \sum\limits_{s=1}^{r_1} \bar{y^d}/y_{sk}^d + \sum\limits_{s=1}^{r_2} \bar{y^u}/y_{qk}^u \right)}$$

$$s.t. \begin{cases} \bar{x} \geq \sum\limits_{j=1,\neq k}^{n} x_{ij}\lambda_j, i = 1,\dots,m; \\ \bar{y^d} \geq \sum\limits_{j=1,\neq k}^{n} y_{sj}^d\lambda_j, s = 1,\dots,r_1; \\ \bar{y^u} \geq \sum\limits_{j=1,\neq k}^{n} y_{qj}^u\lambda_j, q = 1,\dots,r_2; \\ \lambda_j \geq 0, j = 1,\dots,n; \\ \bar{x} \geq x_{ik}, i = 1,\dots,m; \\ \bar{y^d} \leq y_{sk}^d, s = 1,\dots,r_1; \\ \bar{y^u} \geq y_{qk}^u, q = 1,\dots,r_2 \end{cases} \quad (1)$$

Based on Zhang et al. and Guo et al. [53,54], this study selects the GDP of each city as the expected output and the $CO_2$ emissions of each city as the undesired output. The input factors are fixed asset investment, employment, and energy consumption. The descriptive statistical characteristics of the input and output variables in each region are listed in Table 2.

**Table 2.** Descriptive statistical characteristics of the input and output variables (2003–2017).

| Urban Agglomeration | Index | Fixed Asset Investment (100 Million yuan) | Employment (10,000) | Energy Consumption (10,000 tce) | GDP (100 Million yuan) | CO₂ (10,000 t) |
|---|---|---|---|---|---|---|
| Yangtze River Delta | Maximum | 5262.311 | 1346.709 | 23376.55 | 72.9937 | 57429.17 |
| | Minimum | 34.9052 | 11.7169 | 114.0091 | 30632.99 | 280.0861 |
| | Mean | 1271.943 | 177.7579 | 3803.852 | 2619.365 | 9344.922 |
| | Standard deviation | 1122.808 | 201.1752 | 4377.23 | 3354.419 | 10,753.54 |
| Middle reaches of the Yangtze River | Maximum | 12640 | 448.3031 | 11,433.28 | 13,410.34 | 28,088.13 |
| | Minimum | 29.7305 | 14.33 | 120.3828 | 78.28177 | 295.7444 |
| | Mean | 730.7348 | 86.85066 | 1503.13 | 1066.691 | 3692.738 |
| | Standard deviation | 1036.328 | 70.25417 | 1811.872 | 1281.369 | 4451.225 |
| Chengdu–Chongqing Sum | Maximum | 12318.4 | 1551.44 | 17,703.25 | 19,500.27 | 43,491.58 |
| | Minimum | 0.0076 | 17.45 | 240.3739 | 156.761 | 590.5265 |
| | Mean | 917.5525 | 133.1638 | 1948.825 | 1340.323 | 4787.679 |
| | Standard deviation | 1855.081 | 260.6991 | 3211.276 | 2345.294 | 7889.141 |

### 2.3. Spatial Auto-Correlation Model

According to most studies, a certain attribute value or a certain economic–geographic phenomenon on a region space unit is related to the same phenomenon or attribute value on the adjacent region's space unit. Ignoring the spatial auto-correlation problem may lead to an error in the analysis and estimation of spatial effects. Scholars often use Moran's I to test whether the observations in an area exhibit spatial correlation [55]. Moran's I range is "−1 to +1," and the closer the index is to 1, the greater the degree of spatial positive auto-correlation; conversely, the closer the I value is to −1, the greater the degree of negative spatial correlation. When the I value is equal to 0, it represents that there is no spatial correlation in emission efficiency values. The specific calculation formula is as follows:

$$
\begin{aligned}
\text{Moran's I} &= \frac{n \sum_{i=1}^{n} \sum_{j=1}^{n} W_{ij}(y_i - \overline{y})}{\sum_{i=1}^{n} \sum_{j=1}^{n} W_{ij} \sum_{i=1}^{n} (y_i - \overline{y})} \\
&= \frac{\sum_{i=1}^{n} \sum_{j=1}^{n} W_{ij}(y_i - \overline{y})(y_j - \overline{y})}{S^2 \sum_{i=1}^{n} \sum_{j=1}^{n} W_{ij}}
\end{aligned}
\tag{2}
$$

where, $n$ is the number of spatial units; $y_i$ and $y_j$ is the attribute value of regions I and $j$, respectively; $\overline{y}$ is the average value of the attributes of all regions; $S^2$ is the variance of the attributes; and $W_{ij}$ is the spatial weight matrix. Commonly used spatial weight matrices include adjacency (0–1) spatial weight matrix, geographic-distance spatial weight matrix, and economic–geographic-distance spatial weight matrix. Considering the process of economic development, the degree of impact continues to weaken as the distance increases, because related activities influence carbon emission efficiency in neighboring areas. Concerning the choice of the spatial weight matrix, the actual geographic spatial correlation is considered in the model. Under the spatial weight matrix of geographic distance, the spatial correlation of carbon emission efficiency between cities is more significant. Therefore, this article draws on the method [56]. The formula used to calculate the distance

of each city landmark based on latitude and longitude for use in the development of the weight matrix [57] is as follows:

$$W_{ij} = \begin{cases} \frac{1}{d_{ij}^2}, i \neq j \\ 0, i = j \end{cases} \tag{3}$$

To ensure the accuracy of the global Moran's I index, the Z test needs to be used to further test the spatial significance. The formula is as follows:

$$Z(d) = \frac{MoranI - E(I)}{\sqrt{VAR(I)}} \tag{4}$$

$$E(I) = -\frac{1}{n-1} \tag{5}$$

$$VAR(I) = \frac{n^2 w_1 + n w_2 + 3 w_0^2}{w_0^2 (n^2 - 1)} - E^2(I) \tag{6}$$

*2.4. Spatial Measurement Model*

Through Moran's I auto-correlation test, the carbon emission efficiency of the three urban clusters in the Yangtze River Economic Belt all have significant spatial correlation characteristics. Therefore, it is necessary to comprehensively consider the spatial correlation and spillover effects to ensure the accuracy of the model regression results. Currently, the most widely used spatial models are the spatial lag model (SLM), the SEM, and the SDM. The SLM assumes that the influencing factors of the explanatory variable can be used in other spatial regions by the spatial transmission mechanism; the SEM assumes that the spatial dependence reflects the spatial correlation of the random error term; and the SDM integrates the characteristics of SLM and SEM, and introduces the spatial lag between the explained variable and the explanatory variable, which can better estimate the spatial effect measurement based on panel data. The general form of the spatial effect measurement model is:

$$\begin{cases} Y_t = \rho W Y_t + \beta X_t + \theta W X_t + \gamma_t A + B + \varepsilon_t \\ \varepsilon_t = \delta W \varepsilon_t + e_t, e_t \sim N(0, \sigma^2 I_n) \end{cases} \tag{7}$$

where $Y_t$ is the explained variable of the spatial unit i at time t, that is, the carbon emission efficiency; $X_t$ represents the explanatory variable; $\beta$ is the parameter vector of the explanatory variable to be estimated; $\rho$ represents the spatial auto-regressive coefficient; $\rho W Y_t$ represents the impact of the explained variable from other cities; $W$ represents the space Weight matrix n × n; $\theta W X_t$ represents the spatial lag term of the explanatory variable, in which $\theta$ is the influence coefficient vector; $\gamma_t$ represents the time effect; $B$ represents the individual effect; $\varepsilon_{it}$ is the random error term; $\delta$ is the spatial auto-correlation coefficient of the random error term $\varepsilon_t$; and $e_t$ represents the random error term. If $\delta = 0$, it can be transformed into a SDM; if $\delta = 0$ and $\theta = 0$, it is a SLM; and if $\rho = 0$ and $\theta = 0$, it can be transformed into a SEM. First, we used the ordinary panel mixed-model LM test to determine the existence of spatial correlation and selected between SLM and SEM models according to their significance levels; secondly, we verified whether SDM can be reduced to SLM or SEM based on the Wald test; and then, the Hanusman test was used to determine whether the effect was fixed random. Finally, if the LR test rejects the null hypothesis, it proves that the SDM model is more consistent with the model setting.

*2.5. Space Spillover Test*

Anselin proposed a spatial econometric model to be used to measure spillover effects [58]. Many scholars pointed out that when the explanatory variable or dependent variable has a time lag in the model, the parameters of the model should be explicitly explained [59,60]. Lesage and Page also proposed that using one or more spatial regression models to test the spatial spillover effect leads to certain deviations in the conclusions [61],

and they proposed methods to use partial differential measures to explain the impact of the variables. The modified SDM equation is as follows:

$$Y_t = (I - \rho W)^{-1} + (I - \rho W)^{-1}(X_t\beta + WX_t\theta) + (I - \rho W)^{-1}(\gamma_t A + B) + (I - \rho W)^{-1}\varepsilon_t \tag{8}$$

The partial differential equation matrix of $k$ variables in the dependent variable is as follows:

$$\begin{bmatrix} \frac{\partial Y}{\partial X_{1k}} & \cdots & \frac{\partial Y_n}{\partial X_{1k}} \\ \vdots & \vdots & \vdots \\ \frac{\partial Y_n}{\partial X_{1k}} & \cdots & \frac{\partial Y_n}{\partial X_{nk}} \end{bmatrix} = [(I - \rho W)]^{-1} \begin{bmatrix} \beta_k & W_{12}\theta_k & \cdots & W_{1n}\theta_k \\ W_{21}\theta_k & \beta_k & & W_{2n}\theta_k \\ \vdots & \vdots & \vdots & \vdots \\ W_{n1}\theta_k & W_{12}\theta_k & \cdots & \beta_k \end{bmatrix} \tag{9}$$

In partial differential equations, the mean value of the diagonal elements represents the direct spatial effect, and the mean value of the row sum or column sum of the off-diagonal elements represents the spatial spillover effect.

## 3. Empirical Results and Discussion

### 3.1. Analysis of Carbon Emission Efficiency and Regional Differences in Urban Agglomerations

We use the superefficiency SBM model that considers undesired output to analyze the average carbon emission efficiency of the three urban clusters in the Yangtze River Economic Belt and their inner cities and to obtain the comprehensive carbon emission efficiency of the three urban agglomerations and 69 cities based on Max DEAv6.3. Value, pure technical efficiency value, scale efficiency value, and TE = PTE × SE are shown in Table 3.

At the urban agglomeration level, only the average carbon emission efficiency of the middle reaches of the Yangtze River has reached the effective production frontier, and the comprehensive carbon emission efficiency is 1.12. The pollution in the other two urban clusters has exceeded the environmental carrying capacity, and the energy conservation and emission reduction efficiency are low. Among them, the comprehensive efficiency values of the Yangtze River Delta urban agglomeration and Chengdu–Chongqing urban agglomeration are 0.99 and 0.62, respectively. It can be seen from Table 3 that the scale efficiency of the three urban agglomerations has not reached scale efficiency. The Yangtze River Delta urban agglomeration has the highest scale efficiency value of 0.93, and the Chengdu–Chongqing urban agglomeration has a scale efficiency value of only 0.81, showing a decreasing trend in the east, middle, and west. It is consistent with the research based on the level of carbon emission efficiency at the provincial level in my country.

It can be seen from Table 3 that there are significant differences in the carbon emission efficiency across cities. Among the city clusters in the Yangtze River Delta, only one city in Taizhou has achieved effective energy-saving and emission reduction efficiency. Its pure technical efficiency and scale efficiency values are 1.01 and 0.99, respectively; Chizhou has the lowest efficiency value at only 0.39, with a 61% scale efficiency, which has become a key factor limiting its efficiency improvement. Among the urban agglomerations in the middle reaches of the Yangtze River, Huanggang, Jingmen, Xiaogan, Xiangyang, and Xiangtan have higher carbon emission efficiency values, with an average value > 0.8. However, Yingtan, Xinyu, Yichun, Pingxiang, Shangrao, Fuzhou, and Jian have lower carbon emission efficiency values, with all the average values being lower than 0.5, and there is still much room for improvement in pure technical efficiency. Compared with the former two, the overall performance of the Chengdu–Chongqing urban agglomeration is the worst. No city has an efficiency value of 1, which is below the production frontier. The average efficiency value of Deyang City is 0.9, and its pure technical efficiency value and scale efficiency are 0.98 and 0.92, respectively, which makes it the city with the highest overall efficiency value in the Chengdu–Chongqing urban cluster. Meanwhile, Chongqing, the least efficient city, has an average efficiency value of only 0.41. There is much room for improvement in both pure technical efficiency and scale efficiency. This also illustrates the arduous nature

of the task of energy conservation and emission reduction in the Chengdu–Chongqing urban agglomeration.

**Table 3.** Average carbon emission efficiency value of urban clusters and its decomposition (2003–2017).

| Urban Agglomeration | City | TE | PTE | SE | SR | City | TE | PTE | SE | SR |
|---|---|---|---|---|---|---|---|---|---|---|---|
| Yangtze River Delta City Group | Shanghai | 0.92 | 1.22 | 0.75 | − | Huzhou | 0.61 | 0.63 | 0.96 | + |
| | Nanjing | 0.55 | 0.62 | 0.88 | − | Saoxing | 0.90 | 0.93 | 0.97 | − |
| | Wuxi | 0.86 | 0.99 | 0.87 | − | Jinhua | 0.90 | 0.96 | 0.94 | + |
| | Changzhou | 0.57 | 0.6 | 0.95 | − | Zhoushan | 0.72 | 0.78 | 0.93 | + |
| | Suzhou | 0.66 | 0.81 | 0.82 | − | Taizhou | 1.00 | 1.01 | 0.99 | − |
| | Nantong | 0.79 | 0.86 | 0.92 | − | Hefei | 0.33 | 0.34 | 0.98 | + |
| | Yancheng | 0.72 | 0.76 | 0.95 | − | Wuhu | 0.52 | 0.55 | 0.95 | + |
| | Yangzhou | 0.63 | 0.65 | 0.97 | + | Maanshan | 0.49 | 0.64 | 0.77 | + |
| | Zhenjiang | 0.92 | 0.94 | 0.98 | + | Tongling | 0.46 | 0.86 | 0.54 | + |
| | Taizhou | 0.62 | 0.63 | 0.98 | + | Anqing | 0.60 | 0.65 | 0.93 | + |
| | Hangzhou | 0.65 | 0.75 | 0.87 | − | Chuzhou | 0.79 | 0.87 | 0.91 | + |
| | Ningbo | 0.65 | 0.75 | 0.87 | − | Chizhou | 0.39 | 1.21 | 0.32 | + |
| | Jiaxing | 0.60 | 0.63 | 0.96 | + | Xuancheng | 0.60 | 0.79 | 0.77 | + |
| Middle reaches of Yangtze River's urban agglomeration | Wuhan | 0.67 | 0.86 | 0.77 | − | Yiyang | 0.74 | 0.87 | 0.85 | + |
| | Huangshi | 0.70 | 0.78 | 0.90 | + | Changde | 0.79 | 0.84 | 0.94 | + |
| | Ezhou | 0.62 | 1.08 | 0.57 | + | Hengyang | 0.63 | 0.66 | 0.96 | + |
| | Huanggang | 1.01 | 1.02 | 0.99 | − | Loudi | 0.52 | 0.68 | 0.79 | + |
| | Xiaogan | 0.84 | 0.89 | 0.94 | + | Nanchang | 0.58 | 0.62 | 0.93 | + |
| | Xianning | 0.61 | 0.80 | 0.76 | + | Jiujiang | 0.50 | 0.53 | 0.95 | + |
| | Xiangyang | 0.82 | 0.84 | 0.98 | + | Jingdezhen | 0.60 | 0.90 | 0.67 | + |
| | Yichang | 0.69 | 0.70 | 0.99 | + | Yingtan | 0.43 | 1.07 | 0.40 | + |
| | Jingzhou | 0.74 | 0.77 | 0.96 | + | Xinyu | 0.38 | 0.65 | 0.58 | + |
| | Jingmen | 0.96 | 1.00 | 0.96 | + | Yichun | 0.48 | 0.55 | 0.88 | + |
| | Changsha | 0.54 | 0.60 | 0.90 | − | Pingxiang | 0.43 | 0.60 | 0.72 | + |
| | Zhuzhou | 0.61 | 0.64 | 0.96 | + | Shangrao | 0.42 | 0.47 | 0.90 | + |
| | Xiangtan | 0.84 | 0.87 | 0.96 | + | Fuzhou | 0.39 | 0.51 | 0.76 | + |
| | Yueyang | 0.69 | 0.71 | 0.97 | + | Jian | 0.48 | 0.57 | 0.85 | + |
| Chengdu–Chongqing city Group | Chengdu | 0.64 | 0.82 | 0.78 | + | Ziyang | 0.72 | 0.95 | 0.76 | + |
| | Mianyang | 0.81 | 0.87 | 0.93 | + | Zigong | 0.70 | 1.08 | 0.65 | + |
| | Deyang | 0.90 | 0.98 | 0.92 | + | Yibin | 0.59 | 0.65 | 0.91 | + |
| | Leshan | 0.47 | 0.56 | 0.84 | + | Guangan | 0.64 | 0.86 | 0.74 | + |
| | Meishan | 0.58 | 0.80 | 0.72 | + | Dazhou | 0.53 | 0.60 | 0.88 | + |
| | Suining | 0.56 | 0.78 | 0.72 | + | Luzhou | 0.56 | 0.66 | 0.85 | + |
| | Neijiang | 0.65 | 0.85 | 0.76 | + | Chongqing | 0.41 | 0.51 | 0.81 | − |
| | Nanchong | 0.47 | 0.53 | 0.88 | + | | | | | |
| Unit | Yangtze River Delta | 0.99 | 1.06 | 0.93 | + | Chengdu–Chongqing | 0.62 | 0.77 | 0.81 | + |
| | Middle Reaches of The Yangtze River | 1.12 | 1.24 | 0.90 | + | | | | | |

Note: TE is technical efficiency, PTE is pure technical efficiency, SE is scale efficiency, and SR is return to scale change.

### 3.2. Spatial Correlation Analysis of Carbon Emission Efficiency of Urban Agglomerations

3.2.1. Global Autocorrelation Test

According to Equations (2)–(6), based on the spatial geographic distance matrix and Geoda software, the Moran's I value of the carbon emission efficiency of the three urban agglomerations in the Yangtze River Economic Belt from 2003 to 2017 was calculated (Table 4). The Moran's I value of the carbon emission efficiency of the Yangtze River Delta urban agglomeration passed the 10% significance level test and was positive, and the Moran's I value showed a slowly increasing trend, which illustrates the urban agglomeration of the Yangtze River Delta. The carbon emission efficiency of the city exhibits a spatial agglomeration and correlation. The Moran's I value of the carbon emission efficiency of the middle reaches of the Yangtze River urban agglomeration passed the significance test at the 5% significance level. The Moran's I value was the highest among the three urban agglomerations in the Yangtze River Economic Belt, which shows that the carbon emission efficiency of the urban agglomerations in the middle reaches of the Yangtze River shows a strong positive spatial correlation. The improvement in the carbon emission efficiency of the

urban agglomerations is considerably affected by spatial correlation factors; the Moran's I value of the carbon emission efficiency in the Chengdu–Chongqing urban agglomeration in recent years has also shown an increasing trend at a significant level of 10%, and all are positive. Therefore, the carbon emission efficiency within the urban agglomeration of the Yangtze River Economic Belt is not randomly distributed. The spatial distribution pattern generally shows strong spatial agglomeration; namely, regions with similar carbon emission efficiency show significant spatial-clustering characteristics. Consequently, it is necessary to pay full attention to the potential spatial correlation among regions.

**Table 4.** Global auto-correlation of carbon emission efficiency of urban agglomerations in Moran's I (2003–1997).

| Year | Yangtze River Delta City Group | | | Middle reaches of Yangtze River's Urban Agglomeration | | | Chengdu–Chongqing City Group | | |
|---|---|---|---|---|---|---|---|---|---|
| | Moran's I | Z | P | Moran's I | Z | P | Moran's I | Z | P |
| 2003 | 0.164 | 2.244 | 0.034 ** | 0.27 | 2.941 | 0.002 *** | 0.145 | 1.823 | 0.034 ** |
| 2004 | 0.197 | 2.76 | 0.031 ** | 0.288 | 3.127 | 0.001 *** | 0.162 | 2.022 | 0.022 ** |
| 2005 | 0.144 | 1.933 | 0.075 * | 0.338 | 3.528 | 0.000 *** | 0.092 | 1.441 | 0.075 * |
| 2006 | 0.171 | 1.623 | 0.011 ** | 0.342 | 3.564 | 0.000 *** | 0.124 | 1.658 | 0.005 *** |
| 2007 | 0.195 | 2.436 | 0.066 * | 0.373 | 3.86 | 0.000 *** | 0.185 | 2.233 | 0.013 ** |
| 2008 | 0.195 | 2.436 | 0.066 ** | 0.265 | 3.258 | 0.002 *** | 0.159 | 1.976 | 0.046 ** |
| 2009 | 0.221 | 2.941 | 0.002 *** | 0.324 | 3.385 | 0.000 *** | 0.158 | 1.942 | 0.026 ** |
| 2010 | 0.152 | 1.991 | 0.037 ** | 0.239 | 2.579 | 0.005 *** | 0.169 | 2.023 | 0.022 ** |
| 2011 | 0.165 | 1.759 | 0.012 ** | 0.261 | 2.914 | 0.028 ** | 0.182 | 2.849 | 0.002 *** |
| 2012 | 0.195 | 2.436 | 0.066 * | 0.258 | 2.883 | 0.030 ** | 0.153 | 1.984 | 0.024 ** |
| 2013 | 0.163 | 1.641 | 0.127 | 0.315 | 2.842 | 0.030 ** | 0.187 | 2.37 | 0.009 *** |
| 2014 | 0.192 | 2.575 | 0.043 ** | 0.218 | 2.446 | 0.007 *** | 0.241 | 2.837 | 0.002 *** |
| 2015 | 0.193 | 2.642 | 0.044 ** | 0.317 | 3.172 | 0.024 ** | 0.27 | 3.004 | 0.001 *** |
| 2016 | 0.197 | 2.76 | 0.039 ** | 0.347 | 3.762 | 0.039 ** | 0.219 | 2.234 | 0.080 * |
| 2017 | 0.194 | 2.644 | 0.042 ** | 0.341 | 3.713 | 0.030 ** | 0.204 | 2.133 | 0.003 *** |

Note: ***, **, and * indicate the significance level of 1%, 5%, and 10% respectively.

### 3.2.2. Local Correlation Test

To further test the spatial relationship of the carbon emission efficiency of urban agglomeration and investigate the distribution characteristics of the carbon emission efficiency of urban agglomeration, the article draws the local Moran index scatter plot of three urban agglomerations based on the spatial weight matrix of geographic distance, as shown in Figures 1–3.

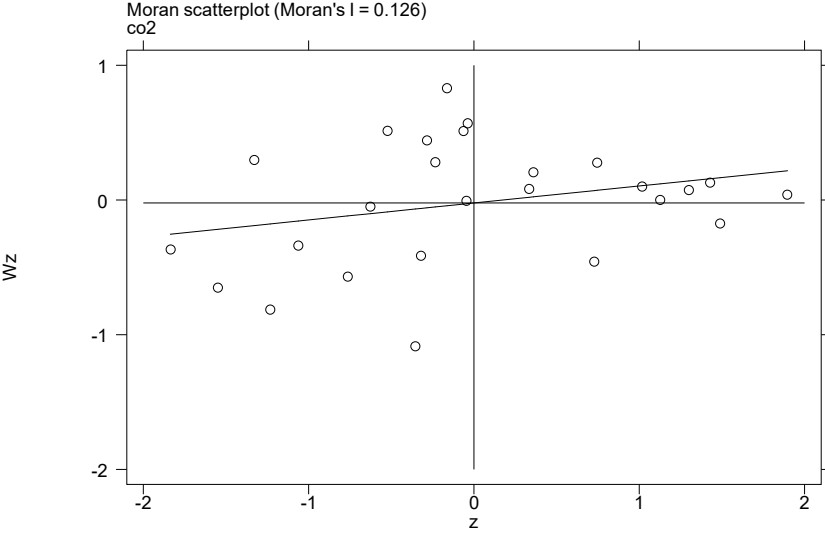

**Figure 1.** Moran's I scatter plot of carbon emission efficiency in the Yangtze River Delta City Group from 2003 to 2017.

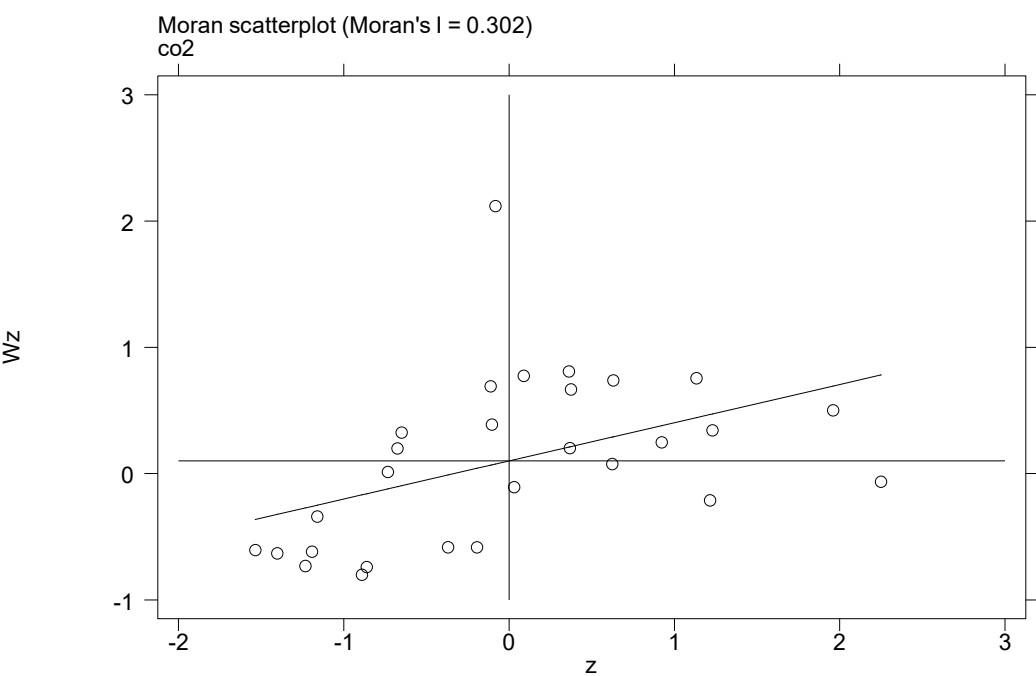

**Figure 2.** Moran's I scatter plot of carbon emission efficiency in middle reaches of Yangtze River's urban agglomeration from 2003 to 2017.

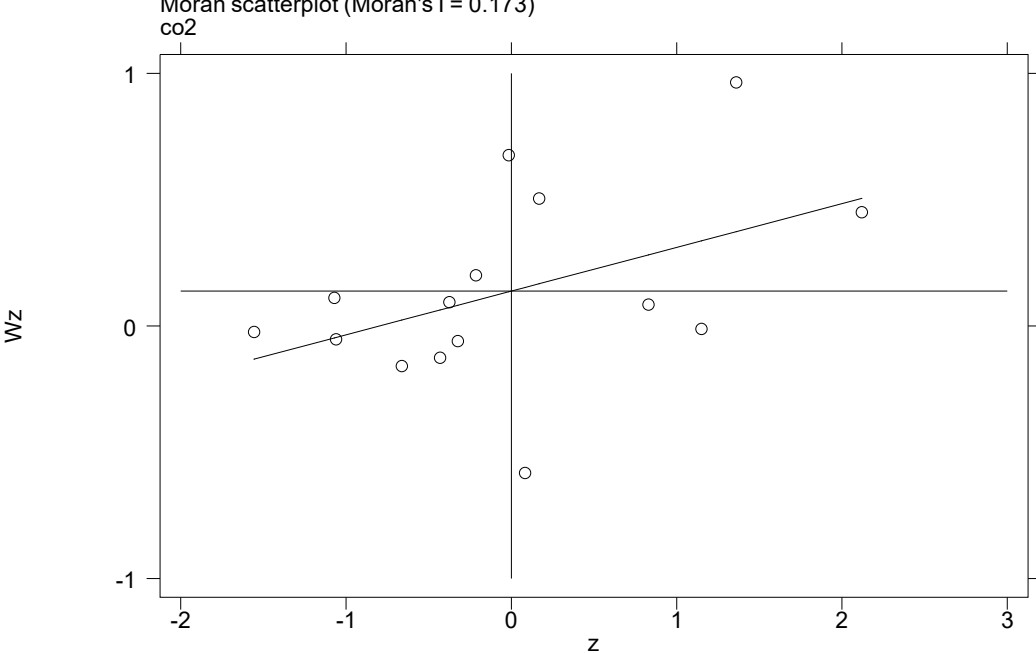

**Figure 3.** Moran's I scatter plot of carbon emission efficiency in Chengdu–Chongqing city Group from 2003 to 2017.

Figure 1 is a Moran's I scatter chart of the average carbon emission efficiency of the Yangtze River Delta urban agglomeration from 2003 to 2017. The carbon emission efficiency of the Yangtze River Delta urban agglomeration not only exhibits significant spatial auto-correlation but also has a certain level of spatial heterogeneity: namely, most cities are in spatial clusters, and a few cities are in spatial outlines. Among them, only three cities in the local area of the urban agglomeration's Moran's I index passed the 10% significance level test. As the core city of the Yangtze River Delta urban cluster, Hefei belongs to the local auto-correlation low–low (L–L) type area. Cities such as Chuzhou and Ma'anshan have

formed regions in which carbon emission efficiency is concentrated. Tongling and Chizhou, located in Hefei and southern Anhui Province, also have L–L correlations. Secondly, 42% of the cities in the urban clusters are located in the second and fourth quadrants, so that low-efficiency cities are adjacent to high-efficiency cities, indicating a negative spatial distribution of carbon emission efficiency in urban clusters. The surrounding urban agglomerations, dominated by central cities, such as Shanghai, Hefei, and Wuxi, have a more positive spatial auto-correlation. According to Figure 2, nearly 71% of the cities in the middle reaches of the Yangtze River had obvious positive spatial auto-correlations. The 10 cities of Xiangyang and Jingzhou belong to the high–high (HH) type, and the nine L–L cities are mainly distributed in Jiujiang and Yingtan in the north and east of the middle reaches of the Yangtze River. The other nine cities are in H–L and L–H areas, concentrated in the southwest of the city centered on Changsha. From a local perspective, nine cities in the urban agglomeration have passed the 10% significance level test. Among them, the three cities of Xiangyang, Jingzhou, and Jingmen are away from the city center in the northwestern area of the middle reaches of the Yangtze River, which belong to local auto-correlation H–H type areas. Meanwhile, six cities, including Yingtan and Fuzhou, which are in the southeast of the urban cluster, belong to the local auto-correlation L–L type area. According to Figure 3, 80% of the cities had positive spatial relevance. Among them, eight cities are in L–L areas, three cities are located in H–H areas, and they belong to L–H and H–L types, respectively, with different spaces. Cities with different spatial auto-correlations accounted for only 20% of the cities. In an internal area, only two cities, Mianyang and Deyang, had a Moran's I index that passed the 10% significance level test, and both cities are located in the northwestern part of the Chengdu–Chongqing urban agglomeration and belong to local H–H areas. The above results show that the spatial correlation and spatial agglomeration of carbon emission efficiency of the urban agglomerations in the Yangtze River Economic Belt are apparent.

### 3.3. Influencing Factors and Method Selection

#### 3.3.1. Urbanization and FDI Control Variable Indicators

Due to the impact of urbanization and foreign investment on carbon emission efficiency in the process of the Yangtze River Economic Belt urban agglomeration, we refer to the research results of domestic and foreign scholars on the quality of urbanization and FDI on carbon emission efficiency [62,63]. Based on the panel data of three urban agglomerations in the Yangtze River Economic Belt from 2003 to 2017 and considering the multi-dimensionality and comprehensiveness of the indicators, the indicators selected in this article are as follows: towns' urbanization rate (UR), economic prosperity (EA), industrial structure (IF), population density (PD), and energy intensity (EI) to represent the level of regional urbanization. Considering the influence of urbanization development on carbon emission efficiency, in addition to urban population indicators, analyses should also include factors such as economic development, industrial structure distribution, resource consumption patterns, and urban life indicators, as well as the overall impact of urbanization level on carbon emission efficiency. To compare the level of foreign investment in various cities, FDI as a percentage of GDP is selected to represent the degree of openness. The original data are from the "China City Statistical Yearbook", "China Energy Statistical Yearbook", and provincial statistical yearbooks. The specific variables are described in Table 5.

#### 3.3.2. The Choice of Spatial Effect Measurement Model

From the test results in Table 6, it can be seen that during the study period, the values of Moran's I in the three urban agglomerations in the Yangtze River Economic Zone passed the 1% level significance test, and the spatial correlation of carbon emission efficiency is also apparent. On this basis, a spatial measurement model was used. First, according to the LM test results, it can be seen that LM lag, LM error, R-LM lag, and R-LM error of the three urban agglomerations are all positive, and most urban agglomerations passed the

1% level significance test, that is, the tests confirmed the auto-correlation in the residual space estimated by the model. Secondly, through Wald and LR likelihood ratio tests, it was found that Wald-spatial lag, Wald-spatial error, LR-spatial lag, and LR-spatial error all passed the 1% level significance level test, indicating that the SDM is more suitable than the SLM and the SEM, and there are individual effects and time effects. Finally, the Hausman test found that the three major urban agglomerations all passed the 1% significance level; that is, the data in this article are more suitable for establishing a fixed effects model. In summary, the spatial Doberman model with two-way fixed effects is selected for spatial effects research.

**Table 5.** Descriptive statistics of each variable.

| Variable | Variables' Definition and Unit | Variable Symbol | Unit | Maximum | Minimum | Mean | Standard Deviation |
|---|---|---|---|---|---|---|---|
| Urbanization | Urban population as a percentage of total population | UR | % | 89.6 | 0.82 | 50.9079 | 13.7375 |
| | GDP per capital | EA | 10,000/per | 250,644.1 | 3383.506 | 13.7375 | 29609.67 |
| | Percentage of added value of secondary industry to GDP | IF | % | 74.73 | 24.48 | 50.347 | 7.733 |
| | Number of people living on land per unit area | PD | People/km$^2$ | 2294.591 | 183.1466 | 551.0727 | 295.8647 |
| | Average total energy consumption per 10,000 yuan | EI | tce/10,000 | 4.5059 | 0.5078 | 1.557908 | 0.5078 |
| Foreign investment level | Foreign direct investment as a percentage of GDP | OP | % | 89.6 | 0.82 | 29.1885 | 3.3040 |

**Table 6.** Spatial dependence tests of the selected urban agglomerations.

| Parameter | | Yangtze River Delta City Group | | Middle Reaches of Yangtze River's Urban Agglomeration | | Chengdu–Chongqing City Group | |
|---|---|---|---|---|---|---|---|
| | | Coefficient | *p*-Value | Coefficient | *p*-Value | Coefficient | *p*-Value |
| Moran's I error term | | 4407.13 | 0.000 | 1475.23 | 0.000 | 13000 | 0.000 |
| LM test | LM lag | 17.92 | 0.000 | 25.27 | 0.000 | 135.43 | 0.000 |
| | R-LM lag | 4.65 | 0.031 | 21.32 | 0.000 | 103.35 | 0.000 |
| | LM error | 41.90 | 0.000 | 4.17 | 0.041 | 32.62 | 0.000 |
| | R-LM error | 28.64 | 0.000 | 0.23 | 0.033 | 0.54 | 0.463 |
| Wald test | Wald-spatial lag | 39.89 | 0.000 | 71.63 | 0.000 | 94.15 | 0.000 |
| | Wald-spatial error | 39.75 | 0.000 | 70.44 | 0.000 | 94.12 | 0.000 |
| LR test | LR-spatial lag | 36.35 | 0.000 | 58.49 | 0.000 | 89.74 | 0.000 |
| | LR-spatial error | 36.14 | 0.000 | 58.42 | 0.000 | 89.53 | 0.000 |
| Hausman test | | 67.04 | 0.000 | 36.25 | 0.000 | 30.97 | 0.000 |
| LnL | | 191.2139 | | 235.9849 | | 314.1120 | |

### 3.4. An Empirical Analysis of the Spatial Effect of Carbon Emission Efficiency

3.4.1. Regression Results of the Spatial Durbin Mode

From the previous test results, it can be seen that the spatial Durbin model cannot be reduced to a post-space model or a SEM. Therefore, to comprehensively analyze the spatial effects of urbanization and FDI on the emission efficiency of the Yangtze River Economic Belt urban agglomeration, the interception uses the fixed effect SDM that includes the spatial weights of the dependent and independent variables for empirical analysis.

Based on Table 7, both EA and PD passed the significance level test and the coefficients are positive. Namely, the two factors promote carbon emission efficiency in the urban agglomerations in the Yangtze River Economic Belt, and the per capita GDP has a greater impact than PD. In contrast, IF, logarithm of energy intensity (LnEI), and OP are all negative at different significance levels, indicating that the three factors would improve carbon emission efficiency to a certain degree. Among them, LnEI has the least impact, followed by IF, and FDI has the greatest impact. In addition, UR had a significant negative effect on carbon emission efficiency in the middle reaches of the Yangtze River and the

Chengdu–Chongqing urban agglomeration. Although the coefficient of the Yangtze River Delta urban agglomeration is positive, the result is not significant.

**Table 7.** Regression results of the spatial Durbin model.

| Variable | Yangtze River Delta City Group | Middle Reaches of Yangtze River's Urban Agglomeration | Chengdu−Chongqing City Group |
|---|---|---|---|
| UR | 0.0271409 (0.25) | −0.200226 *(−1.28) | −0.8568345 *** (5.41) |
| EA | 0.0435023 * (1.50) | 0.1417814 ***(5.39) | 0.3217309 *** (6.73) |
| IF | −0.1306363 * (−1.43) | −0.5168235 *** (−4.55) | −0.2002263 * (−1.78) |
| LnPD | 0.0382642 *(1.65) | 0.0217736 * (1.20) | 0.1016655 *** (5.41) |
| LnEI | −0.2033711 *** (−10.89) | −0.1615388 *** (12.41) | −0.1610863 *** (−3.65) |
| OP | −0.321099 * (−1.42) | −0.6500517 *** (−3.03) | −2.216825 *** (−2.96) |
| W*UR | 0.0630485 (0.19) | 0.4694512 * (1.22) | 4.140133 *** (4.98) |
| W*EA | 0.262339 *** (3.83) | 0.2379758 *** (3.88) | −0.2159406 * (−1.60) |
| W*IF | −0.4736788 * (−1.32) | −0.935617 *** (−3.15) | 0.5361939 * (1.59) |
| W*LnPD | −0.0546346 (−0.77) | −0.0436381 * (−1.03) | 0.717352 *** (5.66) |
| W*LnEI | 0.1070017 * (1.62) | −0.0818094 ** (−1.95) | −0.4925147 *** (−3.50) |
| W*OP | 1.26773 * (1.78) | −4.095381 *** (−5.30) | −1.240943(−0.45) |
| $R^2$ | 0.1922 | 0.2070 | 0.153 |
| $\rho$ | 0.215943 *** (13.78) | 0.12876 *** (14.41) | 0.06959 *** (10.68) |
| Log-likelihood | 191.2139 | 314.1120 | 235.9849 |

Note: Numbers inside the () represent t-stat values; ***, **, and * are significant at the levels of 1%, 5%, and 10%, respectively; the z statistic value is in brackets; W represents the spatial weight matrix; and the meanings of other variables and parameter symbols are the same as in Table 5.

The reason lies in the development of the Yangtze River Economic Belt: coal consumption in the Yangtze River Economic Zone is as high as 70% of the total consumption, while renewable resources such as hydropower and solar energy account for only 10%. This is also an important reason for the excessive carbon emissions in the entire region. Meanwhile, the secondary industry is mainly used through steel. The IF model of coal and other energy-intensive industries driving GDP growth also increases carbon emissions in the Yangtze River Economic Zone; in addition, FDI has not only driven economic growth but also promoted investment. The transfer of domestic energy development, processing, and other industries with strong energy dependence, severe environmental pollution, and low added-value result in low-carbon emission efficiency in the entire region. However, the improvement in the level of economic development has also gradually increased the contribution rate of carbon emission reduction to a certain extent. Adequate funds to introduce low-carbon technology and equipment facilitate the improvement of energy efficiency and reduce carbon emission efficiency. Travel behavior preferences, factors influencing them, and carbon emission reduction potential could be used to formulate carbon emission reduction policy recommendations that promote urban residents to share travel.

Excluding the insignificant W*UR and W*lnPD coefficients of the Yangtze River Delta city group, the other two city groups passed the significance test; the three city groups, W*EA, W*IF, and W*LnEI, all passed the significance level test. Regarding FDI, except for the W*OP of the Chengdu–Chongqing urban agglomeration, which failed the significance test, the W*OP of the other two urban agglomerations were significantly correlated. There was a significant spatial spillover effect between the spatial lag term of the dependent variable and the spatial interaction term of the independent variable in each urban agglomeration; namely, urbanization and FDI factors improve or decrease the carbon emission efficiency of neighboring cities in space, which further verifies the spatial spillover effects of various factors on the carbon emission efficiency of the urban agglomeration in the Yangtze River Economic Belt.

3.4.2. Spatial Spillover Estimation Results

Based on the SDM, the total effects of urbanization and FDI on carbon emission efficiency are further decomposed into direct and indirect effects (Table 8). Among them,

the direct effect reflects the impact of urbanization and FDI on the carbon emission efficiency of the region; the indirect effect reflects the spatial spillover effect of urbanization and FDI on the carbon emission efficiency of neighboring areas; and the total effect of urbanization and FDI is the sum of the direct and indirect effects.

**Table 8.** Direct effect, indirect effect, and total effect of spatial Durbin model.

| Variable | Yangtze River Delta City Group | | | Middle Reaches of Yangtze River's Urban Agglomeration | | | Chengdu–Chongqing City Group | | |
|---|---|---|---|---|---|---|---|---|---|
| | Direct Effect | Indirect Effect | Total Effect | Direct Effect | Indirect Effect | Total Effect | Direct Effect | Indirect Effect | Total Effect |
| UR | 0.021 (−0.88) | 0.041 (0.15) | 0.062 (0.24) | −0.400 *** (−2.75) | 0.498 * (1.88) | 0.098 * (1.29) | −1.099 *** (−4.92) | 3.584 *** (5.38) | 2.485 *** (3.29) |
| EA | 0.032 * (1.47) | 0.190 *** (3.50) | 0.222 *** (5.22) | 0.132 *** (5.03) | 0.158 *** (3.11) | 0.29 *** (5.69) | 0.341 *** (7.18) | −0.265 *** (−3.10) | 0.076 * (1.76) |
| IF | −0.106 (−0.88) | −0.331 (−1.15) | −0.437 (−1.43) | −0.468 *** (−3.97) | −0.639 *** (−2.71) | −1.107 *** (−4.37) | −0.226 * (1.56) | −0.473 * (1.56) | −0.699 * (1.73) |
| LnPD | −0.042 * (1.75) | 0.049 * (−1.86) | 0.007 (−0.12) | −0.025 * (1.50) | 0.040 * (−1.56) | 0.015 * (−1.44) | −0.065 *** (2.98) | 0.556 *** (5.33) | 0.491 *** (5.69) |
| LnEI | −0.211 *** (−11.91) | 0.148 *** (3.02) | −0.063 * (−1.28) | −0.159 *** (−12.09) | −0.024 * (1.76) | −0.183 *** (−5.98) | −0.135 *** (−3.08) | −0.341 *** (−3.10) | −0.476 *** (−4.15) |
| OP | 0.400 * (−1.75) | 1.058 ** (2.04) | 1.458 * (1.28) | 0.439 * (−1.94) | 3.234 *** (−5.12) | 3.673 *** (−6.04) | −2.198 *** (−2.99) | −0.403 (0.18) | −2.601 * (−1.55) |

Note: ***, **, and * are significant at the levels of 1%, 5%, and 10%, respectively; the z statistic value is in brackets.

It can be further verified that the carbon emission efficiency of the three urban agglomerations in the Yangtze River Economic Belt is maintained by urbanization and FDI. In the space spillover effect, the fixed effect space was calculated according to the partial differential Equation (9). The direct effects, indirect effects, and total effects of the Doberman model are shown in Table 8. Among them, UR, EI, OP, EA, IF, and PD clearly show spatial spillover effects.

### 3.5. Results and Discussion

The direct impact of UR on the carbon emission efficiency of the Yangtze River Delta urban agglomeration is positive but not significant. In contrast, the impacts on the other two urban agglomerations are significantly negative at the 1% level, indicating that the Yangtze River Delta urban agglomeration has not been in the process of urbanization. This may be due to the Yangtze River Delta region focusing on urbanization development while considering the economic, social, and ecological environment, actively curbing urban pollution, vigorously developing environmentally friendly industries, and abandoning projects that pollute the environment, thereby effectively suppressing $CO_2$ emissions. However, the middle reaches of the Yangtze River and the Chengdu–Chongqing urban agglomeration during the rapid urbanization stage are limited by the shortage of facilities, single industrial structure, deteriorating environmental conditions, and lagging management.

It is challenging to address the problem of urban development quality and rely solely on resources to promote economic development, which would inevitably cause the agglomeration of energy companies and aggravate the current carbon emissions status. Except for the indirect effect of the Yangtze River Delta urban agglomeration, the urbanization development of neighboring cities in the other two urban agglomerations drives the improvement of carbon emission efficiency in the region. It may be because the flow of human, financial, and material resources among cities has been increasing with the improvement of the level of urbanization and the gradual improvement of intercity transportation infrastructure, and cities have improved through technological transformation, transportation, and industrial transfer. The utilization efficiency of coal resources improves the carbon

emission efficiency. From the perspective of the overall effect, the urbanization level of each city in the urban agglomeration has promoted the improvement of carbon emission efficiency for a long time. However, the impact of the degree of urbanization of the Chengdu–Chongqing urban agglomeration is greater, as it is the most densely populated area in the western region. It is located in the upper reaches of the Yangtze River. As of the end of 2018, the urbanization level was 53.8%. The huge population base is an advantage for the human-capital market in the Chengdu–Chongqing urban agglomeration and overall labor quality and productivity. The increase in urbanization would increase the carbon emission efficiency by 2.48 for every 1% increase.

The increase in the level of economic development has a positive and significant direct, indirect, and total effect on the carbon emission efficiency in the Yangtze River Delta and the middle reaches of the Yangtze River, indicating that with the advancement of urbanization, the government is focusing on economic growth. It also considers environmental benefits and exploits the capital and information integration advantages to employ more resources to improve energy-efficiency technologies, energy-saving technologies, and promote the development and application of renewable energy technologies and GHG emission reduction technologies, thereby reducing carbon emissions. However, the improvement in the level of economic development in the Chengdu–Chongqing urban agglomeration has only direct and total effects on carbon emission efficiency and does not facilitate carbon emission reduction in neighboring areas. This may be because, compared with the Yangtze River Delta and the middle reaches of the Yangtze River, the Chengdu–Chongqing urban agglomeration has a lower degree of industrial synergy and lack of important node cities. This has led to inadequate industrial division and coordination, uncoordinated infrastructure between Chengdu and Chongqing, and intensified inter-regional conflicts, presenting a state where competition is greater than cooperation.

IF negatively influences carbon emission efficiency of each urban agglomeration; namely, for every 1% increase in the proportion of the secondary industry, the carbon emission efficiency of the Yangtze River Delta, the middle reaches of the Yangtze River, and the Chengdu–Chongqing urban agglomeration decrease by $-0.437$, $-1.107$, and $-0.247$, respectively. This indicates that there are confounding phenomena in the industrial structure of the urban agglomeration in the Yangtze River Economic Belt. At present, China's secondary industries are mostly heavy-polluting processing and manufacturing industries involved in activities such as energy mining, which generally consumes large quantities of energy, and have considerable negative environmental impacts. The sustained industrialization has resulted in several advanced industrial chains based on traditional industries such as petrochemicals, steel, and automobiles in the urban agglomerations in the Yangtze River Economic Belt. The development of heavy industries has also led to a sharp increase in energy consumption and $CO_2$ and other GHG emissions in various regions. The pollution situation is not optimistic.

The direct impact of lnPD on carbon emission efficiency is significantly negative at 1% and 10%, respectively, which is not beneficial to the improvement of carbon emission efficiency. It may be because the increase in PD accelerate the development of urbanization and increases the demand for urban infrastructure construction, which subsequently increases the use of fuel, steel, cement, and other nonenergy sources, leading to an increase in carbon emissions. Conversely, the surge in population leads to a gradual increase in road carrying capacity, and, in turn, increased traffic and challenges such as excessive vehicle exhaust emissions, as well as further deterioration of the urban environment, which is counterproductive with regard to the implementation of energy-saving and emission reduction policies. Regarding the level of indirect effects, the increase in PD in each city has played a vital role in promoting the improvement of carbon emission efficiency. This may be due to the increase in PD that enables the centralized supply of resources in the region and thus, the large-scale effect. Therefore, the problems of economic development management, high cost, waste of resource input, etc., are avoided, and the cities within the urban agglomeration can influence each other during the development process for mutual

benefit and a win–win scenario. The positive spatial spillover effect offsets the negative direct effect. The overall effect of urban agglomeration shows a weak positive effect.

The direct effect of LnEI on carbon emission efficiency is negative, and both are significant at the 1% level, indicating that excessive energy consumption does not take advantage of the regional carbon emission efficiency improvement. Due to the abundant reserves of energy resources, the urban agglomerations in the Yangtze River Economic Belt rely on coal-based energy resources for rapid economic development. In 2015 alone, the extraction of crude oil, kerosene, and other fossil energy resources reached 63%, 39%, and 20% of the country's total extraction. Excessive consumption of energy exacerbated carbon emissions. The low cost of energy use leads the region to be more inclined to develop energy-intensive industries and to use IT to promote the rapid development of the regional economy, which directly leads to a decline in the level of carbon emission efficiency in various regions. Except for the indirect effect of the Yangtze River Delta urban agglomeration, which shows a positive promotion effect at a significant level of 1%, other urban agglomerations have a significant negative effect. Energy consumption in neighboring regions indirectly inhibits the improvement of carbon emission efficiency in the area. From the perspective of the overall effect, the EI of the three urban agglomerations plays a significant negative role. This may lead to the obvious regional differences among urban agglomerations, and the scale effect is only applicable to the region and cannot be promoted on a larger scale, leading to limited EI promotion

The impact of FDI on the carbon emission efficiency of the Yangtze River Delta and the middle reaches of the Yangtze River is significantly positive, indicating that the introduction of foreign capital has a significant spatial spillover effect. Multinational foreign investors bring advanced and efficient low-carbon technology and equipment, which stimulates the economic growth of local and neighboring cities while reducing carbon emissions and improve the overall carbon emission efficiency; however, foreign investment is not conducive to Chengdu and Chongqing. Compared with other urban agglomerations, the improvement of carbon emission efficiency of the cluster is deeply restricted regarding the city scale, number of cities, and economic competitiveness. The backward traffic conditions and imperfect infrastructure construction make it difficult for foreign investors to enter the region to maximize its effectiveness. Secondly, due to the dense population and abundant energy types in the Chengdu–Chongqing urban agglomeration, several industries with high energy consumption, high pollution, and high emissions, such as the chemical and the steel industry may gradually shift westward, thereby reducing the carbon emission efficiency of the entire urban agglomeration.

The results show that the economic growth of the urban agglomeration in the Yangtze River Economic Belt has increased in coordination with the environment in recent years, and remarkable results have been achieved in energy conservation and emission reductions. The central cities, headed by Wuhan, have natural resource advantages, improved innovation efficiency, and maximized economic development, while they are minimizing the negative impacts of environmental pollution. However, the radiation-driving ability of the central city has a greater role in the organization and leadership of the entire region, which reduces the overall efficiency improvement; the carbon emission efficiency of the Chengdu–Chongqing urban cluster is still at a low level, which may be due to long-term factors such as natural geographical conditions, openness to the outside world, technological management differences, and urbanization quality. The economic growth model at the expense of the environment has caused pollution emissions to exceed the environmental carrying capacity, resulting in low efficiency in energy savingss and emission reductions. Compared with the Yangtze River Delta and the middle reaches of the Yangtze River, the abundant and relatively inexpensive natural resources attract many foreign corporations and significant foreign investment. Investors invest in high-pollution and high-energy-consumption industries in the region; therefore, environmental pollution remains high. Only the extensive economic model that relies on increasing the input of hard resources such as labor or capital to achieve economic growth has caused resource

shortages and severe environmental pollution. Overall, the difference in carbon emission efficiency among the three major urban clusters in the Yangtze River Economic Belt is consistent with the development status of each of the urban agglomerations. Among them, the Yangtze River Delta urban agglomeration is one of the two growth poles in China's economic growth. It gathers innovative elements and scientific and technological resources, stimulates vitality in the urban clusters, and enhances the technological spillover effects of the urban clusters. However, the difference in the scale of each city cannot adapt to the economic development in the new era.

In the future, carbon emissions will need to be managed from the source, and carbon emission reduction policies will have to consider local conditions. In addition, regional carbon emissions will have to be reduced through industrial structure adjustment, regional transformation and upgrading, an increase of industrial technology, and promotion of the improvement of the overall carbon emission efficiency of the Yangtze River Economic Belt.

## 4. Conclusions and Suggestions

### 4.1. Conclusions

This study uses the panel data of 69 cities in the three major urban agglomerations of the Yangtze River Economic Belt from 2003 to 2017 to calculate the carbon emission efficiency of the three major urban agglomerations in the Yangtze River Economic Belt based on the superefficiency SBM model and explores spatial dependence under the fixed effect of time. The Durbin model confirmed the spatial dependence of carbon emission efficiency and the spillover effects of urbanization and FDI factors, and we empirically examined the degree of impact of each factor on the carbon emission efficiency of three urban agglomerations in the Yangtze River Economic Belt to draw the following conclusions:

(1) The carbon emission efficiency of urban agglomerations in the Yangtze River Economic Belt differs by region. The urban clusters in the middle reaches of the Yangtze River have the highest efficiencies, followed by the Yangtze River Delta urban agglomerations; the Chengdu–Chongqing urban agglomerations have the lowest carbon emissions efficiency. The overall efficiency is decomposed into direct and indirect effects, and, excluding the Chengdu–Chongqing urban agglomeration, which is greatly affected by pure technical efficiency, the scale efficiency is a key factor inhibiting its improvement in the other two urban agglomerations.

(2) The carbon emission efficiencies of the three urban agglomerations in the Yangtze River Economic Zone all exhibit obvious spatial auto-correlation. The Moran's I scatter diagram shows that the carbon emission efficiencies of the urban agglomerations not only have spatial dependence characteristics but also show degrees of spatial heterogeneity. The carbon emission efficiencies of several cities have obvious H–H and L–L correlations.

(3) Estimations based on the spatial panel measurement model show that levels of urbanization, economic development, and PD all have positive effects on the improvement of carbon emission efficiency, while the industrial structure and EI have negative impacts on carbon emission efficiency.

(4) Based on a foreign investment level perspective, except for the Chengdu–Chongqing urban cluster, which is negatively correlated to FDI and conforms to the "pollution refuge" hypothesis, the other two urban agglomerations have passed the positive significance test and conform to the "pollution halo" hypothesis.

(5) EA and EI are mainly related to the development level of direct carbon emission efficiency through influence. Among them, EA has the strongest direct effect on Chengdu–Chongqing urban agglomeration. UR, IF, PD, and OP are mainly related to the development level of carbon emission efficiency through indirect effects. In particular, OP has a positive spillover effect on the Yangtze River Delta urban agglomeration and the middle reaches of the Yangtze River. On the contrary, it has an unfavorable spillover effect on the Chengdu–Chongqing urban agglomeration. The impact of UR

and IF on the carbon emission efficiency of the Yangtze River Delta city cluster is not obvious.

*4.2. Suggestions*

Based on the above analysis, combined with the basic status quo of the development of the Yangtze River Economic Belt, the following policy recommendations are made:

(1) Coordinate development and promote mutual assistance and cooperation between the upstream, middle, and downstream with green innovation as the driving force and create a high-quality development economic belt. Deploy the leading role of the government and the basic regulation role of the market. There is still much room for improvement in various urban agglomerations with regard to the promotion of energy conservation and emission reduction management, formulation of technical measures, and rules and regulations. In particular, the Chengdu–Chongqing urban cluster should exploit the government's management and supervisory functions and use administrative means to improve energy efficiency and promote regional carbon emission reduction. The city clusters in the middle reaches of the Yangtze River should increase their exploration of carbon rights trading and establish a formal and standardized carbon-trading market to promote energy conservation and emission reduction efficiency. While developing the economy, the city clusters in the Yangtze River Delta should pay attention to the innovation of government functions. Financial support is required to further increase the energy rate to promote the development of green and low-carbon high-tech industries.

(2) Strengthen energy supervision, reduce energy dependence, and promote healthy energy flow. Energy-rich regions are rich in resources and are more inclined to develop heavy industries with high-energy dependence, low added value, and high pollution, such as energy development and processing, and eventually, form an extensive path of development. Conversely, EI also presents a significant spatial spillover effect on the carbon emission efficiency of surrounding areas. Therefore, more attention should be paid to the proliferation of energy endowments, to increase the proportion of alternative energy sources such as hydropower, wind energy, and solar energy, and to reduce the proportion of fossil energy consumption in the Yangtze River Economic Zone, which is dominated by coal, to achieve overall energy conservation and emission reduction.

(3) Actively promote urbanization, improve urbanization quality, and take the road of intensive urbanization. While pursuing the scale and development speed of urbanization, it is also necessary to promote the optimal allocation of the industrial structure, technical structure, and energy factor structure of each urban agglomeration to avoid blind expansion of infrastructure and construction projects. Because of the regional differences in the development level and scale of the urban agglomerations in the Yangtze River Economic Zone, the Yangtze River Delta urban agglomeration should focus on improving the governance of urbanization and orderly control of the scale of urbanization. The urban agglomerations in the middle reaches of the Yangtze River and the Chengdu–Chongqing urban agglomeration should coordinate with more cities and towns. The relationship between urbanization and economic development, rational use of resource advantages, and improvement of energy efficiency strive to achieve both the quality assurance of urbanization and the scale effect of urbanization and ultimately form the transition from an extensive to an intensive development model.

(4) Improve the level of opening-up, unblock trade transmission channels, and accelerate the development of low-carbon trade. Actively introduce high-quality, high-efficiency, and foreign-funded enterprises with advanced industries and green production processes based on geographical advantages. Through the transformation of traditional trade industries, vigorously develop service-oriented trade industries such as green manufacturing and smart manufacturing and restrict foreign investment by increas-

ing taxation. High-energy-consuming industries encourage domestic enterprises and foreign investors to cooperate and exploit the "technology spillover" effect of foreign-funded enterprises in energy conservation and emission reduction. Simultaneously, increase the introduction of clean FDI and guide the distribution of FDI investment areas to maximize the effectiveness of FDI. Conversely, it is necessary to further improve the efficiency of capital use, avoid vicious competition between local governments due to the influx of investments, and reduce economic output growth dependence on capital.

This study attempts to measure $CO_2$ emission efficiency based on an urban agglomeration perspective for the first time. It provides novel insights that could facilitate the study of carbon emission efficiency in the Yangtze River Economic Zone and reveals a series of factors that could influence decision-making units. In addition, the present study takes into account spatial spillover effects, which could improve the accuracy of the results. Nevertheless, there are some limitations. First, due to the relative unavailability of data, the study only covers 2003–2017, and more data is expected to be available in the future to facilitate more comprehensive studies; secondly, this article selects only the two most critical factors, urbanization quality and FDI, to evaluate carbon emissions in urban agglomerations, although factors such as technology, labor, infrastructure development, and other multiword indicators also influence carbon emission efficiency; and in addition, the present study only considers the impact of $CO_2$ pollution on the basin environment, while water, soil, and other resources are critical in the regional environment. Future research activities should consider not only differences among regions but also the influence of multiple and potentially interactive factors on carbon emission efficiency so as to provide more practical policy recommendations for improving carbon emission efficiency in the Yangtze River Economic Zone and to promote coordinated regional development

**Author Contributions:** Conceptualization, S.W. and K.Z.; methodology, S.W. and K.Z.; software, K.Z.; validation, K.Z.; formal analysis, S.W.; investigation, K.Z.; data curation, S.W. and K.Z.; writing—original draft preparation, K.Z.; writing—review and editing, S.W.; visualization, K.Z.; supervision, S.W.; project administration, S.W.; funding acquisition, S.W. All authors have read and agreed to the published version of the manuscript.

**Funding:** This work was supported by the Natural Science Foundation of Shandong Province, China (ZR2019MG030).

**Data Availability Statement:** Data is not publicly available, though the data may be made available on request from the corresponding author.

**Conflicts of Interest:** The authors declare no conflict of interest.

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
