# Peer review of "Influence of Urbanization and Foreign Direct Investment on Carbon Emission Efficiency: Evidence from Urban Clusters in the Yangtze River Economic Belt"

_sustainability, doi:10.3390/su13052722_

Round 1

Reviewer 1 Report

This paper addresses an important issue, and it is based on in-depth investigation of the problem. Although it is informative and generally well-done, it needs improvements in some aspects.

  • Avoid repetition of the word 'investments' in the title.
  • Introduction: I suggest to avoid sub-sections and to state your objective more clearly.
  • Lines 201-202: citation, please!
  • Results MUST be separated from Discussion. This is a standard and strict requirement in all top international journals. Results: only your direct findings. Discussion: generalization of your findings, their further interpretation (how you explain these), comparison to results of the other specialists (also for other countries!), limitations, and policy recommendations (move these to Discussion from Conclusions).
  • Conclusions: please, list 5-7 main, briefly-formulated findings (from Results and Discussion) and state perspectives for further research.
  • The paper needs more citations of articles published in top international journals after 2017 and devoted not only to China. This requirement corresponds to what is suggested to write in Discussion (see above).
  • You MUST indicate data sources and properly cite for all statistics used and shown in this paper (text and tables).

Author Response

Thank you for your comments on our article, the manuscript is titled "Influence of Urbanization and Foreign Direct Investment on Carbon Emission Efficiency: Evidence from Urban Clusters in the Yangtze River Economic Belt", (ID:sustainability-1118153), these comments are revisions and improvements to our paper, and are of great help and value to our paper, and are important to our research instructions. We have carefully studied the 7 comments you put forward and made amendments, hoping to get comments, the main amendments:

Point 1: Avoid repetition of the word 'investments' in the title.

Response 1: Thank you for the title. We have carefully checked and revised the title.

(Lines2-4 page 1)

Point 2: Introduction: I suggest to avoid sub-sections and to state your objective more clearly.

Response 2:Thank you for highlighting this flaw. This section has been revised and revised based on the reviewers’ suggestions, avoiding subsections, and making my goals clearer. (Lines 30-193, page 1、2、3、4).

 Point 3: Lines 201-202: citation, please!

Response 3: Thank you for your advice.We deeply regret our negligence and added a reference to lines 201-202. Please refer to lines 207-209 on page 5 for details.

  • Guo, Q.B.; luo, K. L.; Cheng, C.L. A comparative study on the differences of factors aggregating ability among urban agglomerations in Yangtze River Economic Belt. Progress in Geography. 2020,39(04),542-552.

Point 4: Results MUST be separated from Discussion. This is a standard and strict requirement in all top international journals. Results: only your direct findings. Discussion: generalization of your findings, their further interpretation (how you explain these), comparison to results of the other specialists (also for other countries!), limitations, and policy recommendations (move these to Discussion from Conclusions).

Response 4:We agree with your comment. The results and discussion have been revised based on your opinions. The empirical part of the article is divided into 5 subsections, namely:

3.Empirical Results and Discussion

3.1. Analysis of Carbon Emission Efficiency and Regional Differences in Urban Agglomerations

3.2. Spatial Correlation Analysis of Carbon Emission Efficiency of Urban Agglomerations

3.2.1. Global Autocorrelation Test

3.2.2. Local Correlation Test

3.3. Influencing factors and method selection

3.3.1. Urbanization and FDI control variable indicators

3.3.2. The choice of spatial effect measurement model

3.4. An Empirical Analysis of the Spatial Effect of Carbon Emission Efficiency

3.4.1. Regression Results of the Spatial Durbin Mode

3.4.2. Spatial spillover estimation results

3.5. Results and Discussion

And separate the results of each section from the discussion. For details, please see page 9 line 331-page 23 line 705

Point 5: Conclusions: please, list 5-7 main, briefly-formulated findings (from Results and Discussion) and state perspectives for further research.

 Response 5: Based on your comments, the conclusion section has been further revised. For details, please refer to page 23, lines 706-747.

Point 6:The paper needs more citations of articles published in top international journals after 2017 and devoted not only to China. This requirement corresponds to what is suggested to write in Discussion (see above).

Response 6: Thank you for your comments on the references. This article has added 10 foreign documents after 2017. The details are as follows:

  1. To, A.H.; Ha, D.T.-T.; Nguyen, H.M.; Vo, D.H. The Impact of Foreign Direct Investment on Environment Degradation: Evidence from Emerging Markets in Asia. Int. J. Environ. Res. Public Health. 2019, 16, 1636.

  1. Xie, Q.C.;Wang, X.Y.; Cong, P. How does foreign direct investment affect CO2 emissions in emerging countries? New findings from a nonlinear panel analysis. Journal of Cleaner Production. 2020, 249,119-422. 

  1. Ayamba, E.C.;Haibo, C.; Ibn Musah, A. et al. An empirical model on the impact of foreign direct investment on China’s environmental pollution: analysis based on simultaneous equations. Environ Sci Pollut Res. 2019,26, 16239–16248.

  1. Wang, C.M.;Chu, J. Y. Analyzing on the Impact Mechanism of Foreign Direct Investment(FDI) to Energy Consumption. Energy Procedia. 2019, 159,515-520.

  1. Zhou, Y.; Fu, J.; Kong, Y.; Wu, R. How Foreign Direct Investment Influences Carbon Emissions, Based on the Empirical Analysis of Chinese Urban Data. Sustainability. 2018, 10, 2163.

  1. Ali, R.;Bakhsh, K.; Yasin, M. Impact of urbanization on CO2 emissions in emerging economy: evidence from Pakistan. Sustainable Cities and Society, 2019, 48, 101553. 

  1. Wang, Q.;Wang, The nonlinear effects of population aging, industrial structure, and urbanization on carbon emissions: A panel threshold regression analysis of 137 countries. Journal of Cleaner Production, 2021, 287, 125381. 

  1. Xu, Q.; Dong, Y.X.; Yang, Urbanization impact on carbon emissions in the Pearl River Delta region: Kuznets curve relationships. Journal of Cleaner Production,2018, 180(APR.10), 514-523.  

  1. Wang, S.;Li, C. The impact of urbanization on CO2 emissions in China: an empirical study using 1980–2014 provincial data. Environ Sci Pollut Res. 2018, 25, 2457–2465.

  1. Guo, Q.B.;luo, K. L.; Cheng,L. A comparative study on the differences of factors aggregating ability among urban agglomerations in Yangtze River Economic Belt. Progress in Geography. 2020,39(04),542-552.

Point 7:You MUST indicate data sources and properly cite for all statistics used and shown in this paper (text and tables).

Response 7: Thank you for your comments. The application of all data sources has been re-checked and modified.

Special thanks to you for your good comments.

Reviewer 2 Report

I had the opportunity to complete the second review of this paper. This paper is organized, clear, and the content of the paper is appropriate. There is no padding in the paper. In fact, every paragraph contributes to the argumentation of the paper, and some of the empirical results are tied back to the literature.  The author(s) carefully addressed each of the comments provided in the referee reports.

Author Response

Thank you for your comments on our article, the manuscript is titled "Influence of Urbanization and Foreign Direct Investment on Carbon Emission Efficiency: Evidence from Urban Clusters in the Yangtze River Economic Belt", (ID:sustainability-1118153),We have carefully revised the manuscript according to the reviewers' comments, and also have re-scrutinized to improve the english by a language polishing service.

We hope that the revised manuscript is accepted for publication in the sustainability.

This manuscript is a resubmission of an earlier submission. The following is a list of the peer review reports and author responses from that submission.

Round 1

Reviewer 1 Report

The research topic is interesting and challenging, as the authors address jointly two important drivers of increased CO1 emissions, i.e. FDI and urbanization. I mention below my suggestions for paper improvement:

  • the Introduction is too long and I believe that splitting it into an Introduction - which presents the general research framework, the motivation behind the study, the originality and contributions - and a Literature review is beneficial for readers
  • In the Methodology part, the authors do not properly explain the usefulness of DEA for their research and the specific research objectives that are attained by using DEA/SBM
  • There is no statistical analysis of the data, and the data itself is not presented in the paper. Therefore, one does not know which data is the model applied to.
  • In Table 2, what "TE PTE SE SR" mean? The authors also state that "Value, pure technical efficiency value, scale efficiency value, are shown in Table 2.", but it is not clear where are they present in the table. Moreover, I couldn't link the explanation of the results in Table 2 to the table itself, due to these issues. 
  • Lines 426-427: "We consider the availability of data and 427 its completeness of the initially set six variables." I do not understand what are the authors referring to.
  • At part 3.3.1: the six variables are not statistically described, they are just presented in a table, which is not customary. Then, what does Control Variable Index mean? 
  • I do not understand which is the panel model the authors use, since there is no equation mentioned. This is unacceptable in an academic paper. The fact that the authors mention that one panel specification is better than the other is no replacement for panel model results.
  • What do notations in Table 6 refer to?
  • Overall, the authors fail to explain how the models are effectively used and the link between them and their results. This makes the paper unundertsandable.

Reviewer 2 Report

Comments

In a nutshell:

-    The paper is well written. Good selection and use of reference material. The paper is well-organized and structured with an introduction, a structured body, and a conclusion

1.The authors should interpret and discuss their results based on the relevant research of other scholars, including: 

  • Cai W, Fangyuan T (2020) Spatiotemporal characteristics and driving forces of construction land expansion in Yangtze River economic belt, China. PLoS ONE 15(1): e0227299. https://doi.org/10.1371/journal.pone.0227299
  •   Jianbao Li, Xianjin Huang, Mei-Po Kwan, Hong Yang, Xiaowei Chuai,         Effect of Urbanization on Carbon Dioxide Emissions Efficiency in the         Yangtze River Delta, China, Journal of Cleaner Production (2018), do 10.1016/j.jclepro.2018.03.198
  1. There are typos and grammatical errors. So, the work needs minor language editing.
  2. Another minor critic is about the conclusion, which, although good, could be improved. The conclusion should more explicitly 1) stress the importance of the thesis statement, 2) give the essay a sense of completeness, and 3) leave a final impression on the reader.

I hope that the authors would adequately address the concerns and criticism positively and constructively.

Reviewer 3 Report

I apologize for my statistical and econometrical thought, but please try to write three papers about three homogeneous (sub)populations, and not only one about a compromise called belt

  1. Unclear or false title (including false assumptions about FDI and urbanization as impact and not as factors of intense association with pollution) try to avoid ambiguity from an economic and econometric point of view…
  2. A false assumption like the income is constant during 15 years and GDP/inhabitant is not
  3. Ambiguous results: The paper analysis includes 3 areas or clusters (A, B, C) already ranked, of which C contributes 18% of GDP, 26% of the surface, 29% of the population, 22% of the analyzed relative urbanization (15 cities out of 69) and still with 77% of the average density In the final part, Foreign Direct Investments influences negatively pollution only in C part or C region and positively in A & B regions… I do not know what to think after…please excuse me for that cipollian attitude!

How and Why? In fact, everything looks like an ABC curve in a usual market… with a great difference in concentration-diversification coefficient (if one author evaluates Gini-Struck coefficient, he/she will see that heterogeneity has a maximum value)

  1. In my opinion author(s) could better write 3 papers about spaces like A, B, C, and specific population, and not only one about all three clusters (A, B, C), if they really like a statistical way of thinking or econometrical thought. Total heterogeneity makes completely unclear the entire paper analysis.
  2. I believe the article builds models focused on factors whose R2 or R-squared does not exceed 0.2 in the aggregative or general model, so none of these factors can be significant in 15 years (selected timeline) … I don't think the variables were well analyzed or selected, and I cannot see where is the originality in applying a model or be creative in using false assumptions and hypothesis.

Round 2

Reviewer 1 Report

The authors have made some changes, but, overall, the quality of the paper remains the same. The econometric methodology is not explained (there is no equation, for example, mentioned in the paper) and the opinion I first had, of clumsiness, remains in place.

Reviewer 3 Report

I did not ask for answers. I simply rejected the paper based on five reasons. I received five answers that had nothing in common with my remarks that led me to the same answer or I must refuse the paper again. My only option was to reject the article. Please do not send me this paper the third time...